# Uveitis as an Open Window to Systemic Inflammatory Diseases

**DOI:** 10.3390/jcm10020281

**Published:** 2021-01-14

**Authors:** Thomas El Jammal, Olivier Loria, Yvan Jamilloux, Mathieu Gerfaud-Valentin, Laurent Kodjikian, Pascal Sève

**Affiliations:** 1Department of Internal Medicine, Hôpital de la Croix-Rousse, Université Claude Bernard Lyon I, 69004 Lyon, France; thomas_3901@hotmail.fr (T.E.J.); yvan.jamilloux@chu-lyon.fr (Y.J.); mathieu.gerfaud-valentin@chu-lyon.fr (M.G.-V.); 2Department of Ophthalmology, Hôpital de la Croix-Rousse, Université Claude Bernard Lyon I, 69004 Lyon, France; olivier.loria@chu-lyon.fr (O.L.); laurent.kodjikian@chu-lyon.fr (L.K.); 3Laboratoire UMR-CNRS 5510 Matéis, 69004 Villeurbane, France; 4IMER Department, Hospices Civils de Lyon, 69424 Lyon, France; 5Department of Formation and Research in Human Biology, Université Claude Bernard Lyon 1, HESPER EA 7425, 69008 Lyon, France

**Keywords:** uveitis, sarcoidosis, HLA-B27 associated uveitis, spondyloarthritis, Behçet’s disease

## Abstract

Spondyloarthritis (Spa), Behçet’s disease (BD) and sarcoidosis are major systemic inflammatory diseases worldwide. They are all multisystem pathologies and share a possible ocular involvement, especially uveitis. We hereby describe selected cases who were referred by ophthalmologists to our internal medicine department for unexplained uveitis. Physical examination and/or the use of laboratory and imaging investigations allowed to make a diagnosis of a systemic inflammatory disease in a large proportion of patients. In our tertiary referral center, 75 patients have been diagnosed with Spa (*n* = 20), BD (*n* = 9), or sarcoidosis (*n* = 46) in the last two years. There was a significant delay in the diagnosis of Spa-associated uveitis. Screening strategies using Human Leukocyte Antigen (HLA)-B27 determination and sacroiliac magnetic resonance imaging in patients suffering from chronic low back pain and/or psoriasis helped in the diagnosis. BD’s uveitis affects young people from both sexes and all origins and usually presents with panuveitis and retinal vasculitis. The high proportion of sarcoidosis in our population is explained by the use of chest computed tomography (CT) and 18F-fluorodeoxyglucose positron emission tomography CT that helped to identify smaller hilar or mediastinal involvement and allowed to further investigate those patients, especially in the elderly. Our results confirm how in these sight- and potentially life-threatening diseases a prompt diagnosis is mandatory and benefits from a multidisciplinary approach.

## 1. Introduction

Uveitis is defined as the inflammation of the iris, ciliary body, vitreous, retina, or choroid. Its incidence is 10.5–52/100,000 person–years and the prevalence is 38–284/100,000 persons [1,2,3]. A study of medical insurance claims for 4 million individuals in the USA reported a prevalence of 133/100,000 persons, including a predominance of non-infectious uveitis (90.7%), and anterior uveitis (AU, 80%) [4]. A more recent study, in South Korea, among a national cohort of approximately 1,000,000 Korean residents, found a prevalence of uveitis, AU and non-AU of 17.3, 15.0, and 2.3 per 10,000 persons, respectively [3].

Uveitis is responsible for 5% of legal blindness cases (central visual acuity of 1/10 or less in the better eye), mainly due to macular edema, ocular hypertension, or retinal ischemia [5]. 

Approximately 80 causes of uveitis have been described and these can be classified into five groups including pure ophthalmological entities, infectious [6] and inflammatory diseases, masquerade syndromes, and drug-induced (Table 1). Epidemiology varies according to genetic and ethnic factors (e.g., Human Leukocyte Antigen (HLA)-B27, sarcoidosis, etc.), environmental factors (e.g., tuberculosis in endemic countries), the definition of the disease (e.g., sarcoidosis), the inclusion of certain ophthalmologic entities in the group of idiopathic uveitis (e.g., pars planitis), the paraclinical investigations performed (e.g., nuclear imaging) and the method of patient recruitment (e.g., tertiary centers). This explains the great heterogeneity of the studies reported in the literature. The main etiologies (mostly reported from tertiary centers) are Vogt–Koyanagi–Harada disease (VKH) and sarcoidosis in Japan [7], VKH and Behçet’s disease (BD) in China [8], BD in Turkey [9], HLA-B27-associated uveitis in Australia [10], herpesvirus in North Africa [11] and Thailand [12], tuberculosis in India [13], toxoplasmosis in South America [14], and infections (presumed tuberculosis, followed by cytomegalovirus infection and herpesvirus infection) in Singapore [15]. In Western countries, approximately a quarter of cases are related to ophthalmologic entities, a quarter to systemic diseases meeting consensual diagnostic criteria, a quarter to suspected systemic diseases, and a quarter to unexplained causes [16]. Uveitis of unexplained origin, also known as idiopathic uveitis, accounts for 23% to 44% of cases according to recent studies in the West and Japan [10,17,18]. 

In the last twenty years alone, ten studies have been carried out on the distribution, clinical patterns, and etiologies of uveitis [18,19,20], mostly in ophthalmological centers, in Western Europe. As shown in Table 2, three systemic diseases are particularly prevalent: HLA-B27- and spondyloarthropathy (Spa)-associated uveitis, BD, and sarcoidosis. 

In this study, our attention will focus on these entities.

**Table 1 jcm-10-00281-t001:** Main causes of uveitis in adults (From Sève et al., *Autoimmun. Rev.*, 2017). The most common causes (>0.5%) reported in most recent European series are indicated in bold type [21,22,23,24,25,26,27,28,29].

Etiology	Specific Causes
**Infectious diseases**	Bacterial: **syphilis, tuberculosis**, Lyme disease, cat-scratch disease, rickettsiosis, leptospirosis, brucellosis, Whipple’s disease [30], chlamydiosis, tularemia, post-streptococcal [31] Parasitic: **toxoplasmosis, toxocariasis**, onchocerciasis, cysticercosisViral: **herpes simplex viruses 1 and 2, CMV**, HTLV-1, Dengue virus, Ebola virus, Zika virus, West-Nile virus, Rift valley fever virus, chikungunya virus, coronaviruses [32] Fungal: candidiasis, histoplasmosis, aspergillosis, cryptococcosis
**Inflammatory diseases**	**HLA-B27-associated uveitis/spondyloarthritis** **Chronic inflammatory bowel diseases** **Sarcoidosis** **Behçet’s disease** **Vogt–Koyanagi–Harada disease** **Multiple sclerosis** and anti-myelin oligodentrocyte glycoprotein (anti-MOG)-associated disease [33] **Juvenile idiopathic arthritis** Tubulointerstitial nephritis and uveitis (TINU syndrome) Celiac disease [34] Systemic lupus erythematosus, systemic vasculitides (Kawasaki disease, polyarteritis nodosa, granulomatosis with polyangiitis, giant cell arteritis) Monogenic autoinflammatory diseases: Blau syndrome, cryopyrine-associated periodic syndromes, A20 haploinsufficiency [35] Common variable immunodeficiency [36] IgG4-related disease [37] Kikuchi–Fujimoto disease [38] Sweet’s syndrome [39] Autoimmune lymphoproliferative syndrome [40]
**Pseudo-uveitis**	Trauma, intraocular foreign body Cancer (**oculocerebral lymphoma**, melanoma, retinoblastoma, leukemia, metastasis)
**Ophthalmologic entities**	**Birdshot chorioretinopathy** **Multifocal choroiditis** **Pars planitis** **Fuchs heterochromic cyclitis** Phacoantigenic uveitis Posner–Schlossman syndrome Other white dot syndromes (placoid epitheliopathy, serpiginoid choroiditis) Sympathetic ophthalmia
**Drug-induced uveitis**	Rifabutin Biphosphonates Anti-tumor necrosis factor-α IFN-α or -β BCG therapy Cancer immunotherapy: BRAF and MEK inhibitors, CTLA4 and PD-1/PD-L1 checkpoint inhibitors [41] Vaccines [42]

Abbreviations: BCG: Bacille de Calmette et Guérin; CMV: cytomegalovirus; CTLA4: cytotoxic T lymphocyte antigen 4; HLA: human leukocyte antigen; HTLV-1: Human T-cell Lymphotropic Virus type 1; IFN: interferon; IgG4: immunoglobulin G subtype 4; PD-1: programmed cell death-1; PDL-1: programmed cell death-ligand 1.

**Table 2 jcm-10-00281-t002:** Diagnostic performances of the main biomarkers available in ocular sarcoidosis.

Biomarker	Test Performance/Usefulness			Limitations/Comments	References
Lymphopenia (lymphocyte count <1000/mm^3^) *	Se/Sp	PPV/NPV	YI/AUC	Poor Youden’s index. Test performances based on lymphocyte count cutoff. Better performance coupled with serum ACE.	[43]
	0.15/0.97	0.48/0.85	0.12/0.71		
	Increased sensitivity (0.19), specificity (0.99) and NPV (0.90) when associated with elevated ACE. Increased sensitivity (0.75) but lower specificity (0.77) with 1470/mm^3^ cutoff. Easily accessible, simple, and non-invasive.				
Elevated ACE * (>52–61 UI/l)	Se/Sp	PPV/NPV	YI/AUC	Optimal cutoff varying from 52 UI/l to 61 UI/l. Uninterpretable if patient uses ACE inhibitors.	[43,44]
	0.45–0.78/0.9	0.44/0.89–0.97	0.35–0.68		
	Highly specific and high NPV in patients referred for uveitis with no known cause. Increased sensitivity/specificity coupled with lymphopenia.				
sIL-2R (threshold according to manufacturer)	Se/Sp	PPV/NPV	YI	Not used widely enough (not validated in revised IWOS criteria). No validated threshold.	[45]
	0.98/0.94	0.77/0.99	0.92		
	Very sensitive, high NPV, good YI. sIL2R levels replaced negative tuberculin skin test in Japanese diagnostic criteria for sarcoidosis.				
Lysozyme *	Sensibility: 0.60–78; sensitivity: 76–95. In systemic sarcoidosis, lysozyme levels are positively correlated with sIL-2R levels and ACE levels.			High lysozyme levels can be found in infectious uveitis (tuberculosis, syphilis).	[46,47]
Chitotriosidase activity	No data in sarcoidosis uveitis. In patients with systemic sarcoidosis, 48.8 nmol/h/mL cutoff is associated with 0.89 sensitivity and 0.93 specificity.			High chitotriosidase activity reported in other pulmonary diseases (COPD, asbestosis). Not easily available.	[48]

Abbreviations: ACE: angiotensin converting enzyme; AUC: area under curve; COPD: chronic obstructive pulmonary disease; PPV/NPV: positive and negative predictive value; Se: sensitivity; Sp: specificity; YI: Youden index. * Biomarker included in the IWOS criteria.

Spa encompasses different chronic inflammatory diseases, such as ankylosing spondylitis (AS), reactive arthritis, psoriatic arthritis (PsA), arthritis associated with inflammatory bowel disease (IBD), and undifferentiated spondyloarthritis, which share common clinical features and a strong genetic association with the HLA-B27 antigen. The main symptoms are inflammatory chronic low back pain, peripheral arthritis (typically asymmetric monoarthritis or oligoarthritis predominantly affecting the joints of the lower extremities), dactylitis, and enthesitis [49]. The disease can be complicated by extra-articular manifestations, such as psoriasis, IBD, and acute AU. 

Spa-associated AU is commonly reported as a recurrent unilateral nongranulomatous acute AU and affects approximately one fourth of Spa patients. It is the most common extra-articular manifestation of AS [50]. The risk of uveitis increases with disease duration and appears to be related to a higher cumulative exposure to inflammation [51]. Uveitis is more common in AS (23–33%) than in PsA (7–19%; more frequent in axial PsA than in peripheral disease) or reactive arthritis (26%), IBD (2–5.6%; more often in Crohn’s disease) or non-radiographic axial Spa (15.9%) [52,53,54,55]. A recent large nationwide cohort study in Sweden involving 8517 AS patients, 10,245 undifferentiated Spa (uSpa) and 22,667 PsA showed an incident rate ratios for incident acute AU significantly increased in AS (20.2), uSpa (13.6) and PsA (2.5) compared with controls [56]. Acute AU associated with Spa affects males more frequently than females and typically occurs in young adults (i.e., between 20 and 40) [57,58]. Other ocular manifestations, such as episcleritis and scleritis, may also occur, especially in IBD [57].

BD is an inflammatory disorder characterized by repeated flares of oral and genital ulcers, pustulosis, erythema nodosum, arthritis, and ocular involvement, along with potentially life-threatening vascular, gastrointestinal, and neurological manifestations [59]. BD was first described by Hulusi Behçet, a Turkish dermatologist. This ubiquitous disorder is endemic in Turkey, where the prevalence is the highest (approximately, 80–370 cases/100,000), followed by Iraq, Iran, Korea, and Japan. This population is actually derived from the one present on the ancient Silk Road, from the Mediterranean to the Middle Eastern and Far Eastern countries [60]. Lower prevalence has been reported in North America and northern countries. In the UK, there is an estimated prevalence of 0.64 cases/100,000 [61]. The onset of symptoms typically occurs in early adulthood, but BD is also seen in children and older patients. In the high prevalence areas of Turkey and Middle East, incidence is higher in males. In other countries, sex distribution is variable. The disease is usually severe in young adult men. Several classification criteria exist, but the International Study Group Criteria and International Criteria for Behçet’s Disease are the most widely used [62,63]. Laboratory findings may demonstrate inflammation, including elevation of acute phase reactants. A broad differential diagnosis workup is essential. Ophthalmic involvement, the most debilitating complication of BD, occurs in about 50% of patients, usually as a relapsing-remitting disease [64]. BD uveitis is classically described as an acute, recurrent and nongranulomatous panuveitis associated with occlusive necrotizing retinal vasculitis involving both veins and arteries [65]. Less frequent ocular manifestations include conjunctivitis, conjunctival aphtosis, scleritis, and optic neuritis [66]. 

Sarcoidosis was first described by Besnier et al. in 1889. It is a multi-systemic disease of unknown etiology characterized by the infiltration of various tissues by non-caseating granulomas. Even if the etiology remains unknown, the mechanisms underlying granuloma formation are quite well understood [67]. Sarcoidosis can affect people from all ages and ethnicity but is more frequent in young adults with a later onset in women than in men. About 70% of cases involve patients aged between 25 and 40 years at presentation and a second peak of incidence is observed in women over 50 years old [68]. Its annual incidence is estimated between 2.3 and 11/100,000 [69]. The estimated prevalence varies from 2.17 to 160 cases per 100,000 individuals. This large variability could be explained by the different diagnostic tools used to define sarcoidosis in older series and the varying ethnicity of each cohort. Sarcoidosis course can be divided into two distinct groups: a time-limited course in two-thirds of patients, who evolve through a self-remitting disease within 12 to 36 months [68,69,70], and a chronic course in 10 to 30% of patients who require prolonged treatment. 

Ocular involvement occurs in 25% to 50% of patients with systemic sarcoidosis [71]. Almost every structure of the eye can be affected, and clinical presentation include dry eye, conjunctival granulomas, scleritis, optic neuritis, exophthalmos, and uveitis, which is the most frequent presentation [72,73]. The characteristic signs of sarcoid uveitis are mutton-fat keratic precipitates, Koeppe’s and Busacca’s nodules, trabeculitis, vitreous opacities (i.e., snowballs and snow banks), segmental periphlebitis and microaneurysms on the site of inflamed blood vessels, juxtapapillary and optic disc granulomas, and choroidal nodules of larger proportions [71]. 

These three systemic inflammatory diseases have in common a possible ocular involvement. In many cases, uveitis is the first sign of the disease and the rheumatologist or the internist may play a key role in diagnosing a systemic inflammatory disease. 

We hereby describe selected cases who came to our attention after referral to our ophthalmology department. In all cases the physical examination, the biological and the radiological workup allowed to establish the diagnosis of a systemic inflammatory disease. We have chosen to focus on these three systemic inflammatory diseases and to summarize their main clinical features regarding the ophthalmological presentation, the associated diagnostic approach, and their treatment.

## 2. Experimental Section

### 2.1. Methods

This article is a series of original cases and adheres to the principles of the Declaration of Helsinki of 1964 and its latest amendments. The informed consent of all participants was obtained, and the study was approved by the Local Institutional Review Board. The case reports were chosen among consecutive patients presented to our ophthalmology clinic. Each case report was selected as they were considered by the authors to be either representative of a classical clinical presentation of the disease or representative of the diagnostic issues faced by the clinician. This study received approval from the local ethics committee in February 2019 (No 19–31) and was registered on clinicaltrials.gov (NCT 03863782). In order to describe the updated management of uveitis associated with these diseases, we conducted a review of the English and French medical literature on the Medline database, using the keyword “uveitis”. We selected articles published since 2016, and excluded articles written in languages other than English or French. 

### 2.2. Spondyloarthritis

#### 2.2.1. Case Report

The case of spondyloarthritis presented is that of a 47-year-old man with a history of diabetes mellitus, dyslipidemia, and smoking. In the last three years, he had had several episodes of unilateral AU in both eyes complicated by posterior synechia. He was referred by his ophthalmologist for this uveitis recurring monthly for the past 6 months. Visual acuity was preserved at 0 LogMAR in both eyes, and intraocular pressure was normal. Slit-lamp examination revealed non-granulomatous keratic precipitates with moderate cellular anterior chamber reaction in the left eye, and moderate hypopyon (Figure 1). Fundus examination was normal and optical coherence tomography (OCT) did not reveal macular or papillary edema. The patient also reported inflammatory low back pain, which was relieved by physical exercise. Physical examination of the patient revealed back stiffness. 

Laboratory testing found normal complete blood count (CBC), C-reactive protein (CRP) at 6.5 mg/L, and negative Syphilis serology. Chest X-ray was normal. HLA testing revealed the presence of the HLA-B27 haplotype. Magnetic Resonance Imaging (MRI) of sacroiliac joints found signs of ankylosis in the right side, and major fatty restructuring on both sides, as well as discrete condensation in the anterior part of the articulation, reflecting a history of inflammatory rheumatism. There was no edema in favor of active sacroiliitis (Figure 2). The diagnosis of Spa was established, with a basal activity BASDAI score of 3.5. Treatment for the acute AU included topical steroids with slow tapering, and systemic treatment with sulfasalazine (SSZ) was initiated to decrease the frequency and duration of relapses. AU regressed over the first month and there was only one episode of relapse during the next three years.

#### 2.2.2. Ophthalmic Features and Diagnosis

The association between uveitis and HLA-B27 was first described in 1973, concomitantly with ankylosing spondylitis [74]. In the European Caucasian population, the prevalence of HLA-B27 is 7% and reaches 80% in patients with Spa [75]. The prevalence is lower in Mexicans (4%) and African Americans (2–4%) [76]. Uveitis associated with Spa follows a « rule of 90 »: 87% of cases are unilateral, 91% of cases are anterior, sometimes with reactive anterior vitritis, and 89% are acute [77]. Granulomatous lesions are never found in the anterior segment. In severe forms of acute AU, hypopyon (superimposition of white blood cells in the anterior chamber) can be seen (14–18%) [78]. Uveitis associated with chronic inflammatory bowel disease or psoriatic arthritis, on the other hand, may be associated with anterior granulomatous lesions and tends to be more insidious in the initial phase. It is more often bilateral, chronic and usually involves the posterior segment [53]. The most frequent complications are posterior synechiae (13–90%) [79] and cataract (7–28%) [80]. Ocular hypertension (8–20%), papillitis (2–18%) and cystoid macular edema (6–13%) are less frequent [77]. Recurrences are common (51–67%), with a frequency ranging from 0.6 to 3.3 flares/year. The frequency of relapses tends to decrease over time, and uveitis is more often homolateral [80]. The recurrent nature of these AU is not fully explained yet; possible triggers include infections with Gram-negative bacteria, stress, seasonality, and trauma [81]. Male sex, hypopyon, accelerated ESR [78], or an associated Spa [82] are potential risk factors for frequent relapses of HLA-B27-associated uveitis. The visual prognosis of HLA-B27-associated uveitis is rather favorable since less than 2% of patients become legally blind and less than 5% become visually impaired [81]. The risk factors for visual loss are male gender, the presence of posterior synechiae at onset, the use of corticosteroid-sparing therapy or periocular corticosteroid injections, poor control of ocular inflammation, and a chronic course of the disease [83]. 

HLA-B27 haplotype is present in half of patients with acute AU, which is itself associated with Spa in half of the cases [75]. Testing for the HLA-B27 allele is not useful for establishing the etiological diagnosis of intermediate or posterior uveitis [84]. Only 1% of HLA-B27-positive patients have uveitis, suggesting a pathogenic role of other genetic polymorphisms (e.g., *tumor necrosis factor* (*TNF*), *interleukin-10* (*IL-10*), or the *CYP27B1* gene involved in vitamin D metabolism [85,86,87]). Other susceptibility loci (e.g., *ERAP1*, *NOS2*, *IL23R*, and *MERTK*) have recently been identified in a recent large genome-wide association study [88]. In a series of 175 patients with HLA-B27-associated uveitis, 136 patients were followed for Spa, and for 88 of them (64%) the joint disease had not yet been diagnosed at the time of uveitis onset [80]. Several studies showed a delay of up to 7.9 years between the diagnosis of uveitis and the subsequent diagnosis of AS [89]. We conducted a retrospective study of 102 patients referred for etiologic evaluation after an episode of acute AU [90] and found that 21 patients (20.5%) had Spa and 23 (22.5%) were diagnosed with HLA-B27-associated uveitis. We evaluated the relevance of the criterion “inflammatory back pain” (as defined by the Berlin criteria) in the diagnosis of spondyloarthritis (according to the ASAS criteria). Low back pain was 90.5% sensitive and 75.3% specific in establishing the diagnosis of spondyloarthritis. A diagnostic algorithm (Dublin Uveitis Evaluation Tool: DUET) was developed in 101 patients and validated in 72 other patients. This tool was designed to identify criteria indicating the need for referral to a rheumatologist for patients with acute AU [91]. These criteria were: the onset of low back pain before the age of 45 years and lasting more than 3 months, or joint pain requiring referral to a general practitioner associated with a positive HLA-B27 test or known or clinically observed psoriasis. The DUET algorithm was 96% sensitive and 97% specific for the diagnosis of spondyloarthropathy and the positive and negative odds ratios were 41.5 and 0.03, respectively. Oliveira et al. showed that 3/9 patients with recurrent acute AU and no back pain had sacroiliitis on MRI [92]. Further studies are needed to determine whether a positive MRI of the sacroiliac joints, in the absence of axial symptoms, is a good predictor of Spa development. Overall, these data underline the importance for physicians to consider Spa as a cause of non-granulomatous uveitis, especially in young patients with back pain.

#### 2.2.3. Treatment

The treatment of acute AU associated with Spa is not specific and involves frequent and repeated instillations of topical dexamethasone. In a minority of cases, periocular injections of glucocorticoids are necessary to control the inflammatory process. The few patients who do not respond to this treatment usually progress to chronic uveitis, following which they require treatment with methylprednisolone and, occasionally, methotrexate (MTX) or TNF-α antagonists [75]. The selection of patients eligible for maintenance therapy is based on the frequency of recurrences, the presence of residual abnormalities (which are rare), and the patient preference. The type of maintenance therapy depends on the underlying rheumatic features. In patients with frequent recurrences (at least 2–3/year) and mild rheumatic manifestations, SSZ, which had been evaluated in a small study [93] and in a randomized trial including 22 patients [94], is an option [91]. An open clinical trial on 9 patients also suggested the efficacy of MTX at an initial dose of 7.5 to 20 mg/week to reduce the rate of recurrence of AU in patients who had more than 3 recurrences of AU in the past year [95]. A recent study of the effects of SSZ and MTX on the progression of HLA-B27 relapsing acute AU reported a significant decrease in the rates of relapses in both groups [96]. 

TNF-α antagonists prescribed to treat Spa can also prevent uveitis [97]. An overall analysis of data from four placebo-controlled trials (2 of etanercept and 2 of infliximab) in 655 patients who were given TNF-α antagonists for AS showed that the frequency of AU recurrences was 15.6/100 patient-years in the placebo group and 6.8/100 patient–years in the TNF-α antagonist group (*p* = 0.01) [98]. Recurrences were less frequent with infliximab than with etanercept (3.4 vs. 7.9/100 patients–years, non-significant difference). Several observational studies of patients with Spa and a previous history of acute AU have shown a significant decrease of approximately 50% in the rate of uveitis recurrence with TNF-α antagonists, including infliximab, adalimumab (ADA), certolizumab pegol and golimumab but a lesser effect of the soluble TNF receptor fusion protein etanercept [99,100,101]. In addition, several studies suggested that etanercept may cause inflammatory eye disease, including acute AU and scleritis, in patients without a history of inflammatory joint disease predisposing to uveitis [102,103]. The analysis of a German pharmacovigilance database showed a significant increase in the risk of uveitis with etanercept compared to ADA or infliximab [104]. According to the World Health Organization (WHO) causality assessment system, etanercept may be causally related to uveitis independently from age, gender, and HLA haplotype. This adverse event is rare (<1%) and carries no risk of vision loss. Overall, these data have shown that etanercept is less effective in preventing AU compared to the anti-TNF monoclonal antibodies [105]. However, this difference in efficacy was not considered sufficient by the American and European Rheumatology organizations to recommend a treatment with the latter drugs rather than etanercept in patients with an isolated episode of uveitis. [106]. TNF monoclonal antibodies are recommended over etanercept for patients with frequently recurrent AU [106] or, in our opinion, should be preferred in patients with a previous history of severe uveitis related to HLA-B27. Finally, rheumatologists should be aware that TNF-α antagonists might induce a sarcoid-like reaction that may be associated with the development of uveitis. [107]. Vedolizumab, a humanized murine antibody against integrin α4β7 (a protein involved in the migration of leukocytes to the gastrointestinal tract) is approved for the treatment of IBD. In a database analysis, this treatment was associated with a higher frequency of uveitis than in patients treated with TNF-α antagonists. [108]. 

We recently reported an open-label strategy to prevent the recurrences of HLA-B27-associated AU in 61 of 101 patients (60.4%) with Spa-associated uveitis [109]. The use of SSZ as a first-line treatment for the ophthalmologic indication reduced uveitis recurrences in 82% (23/26) of patients. MTX and TNF-α antagonists were initiated for a rheumatologic indication in 81.8% (9/11) and 100% of patients (13/13), respectively. The annual rate of uveitis relapse decreased significantly in patients on SSZ (0.37 relapses/year versus 2.46 relapses/year at baseline; *p* < 0.001) and MTX (1.54 relapses/year versus 4.17/year; *p* = 0.008). The use of anti-TNF-α agents for ophthalmologic purposes was exceptionally necessary (2 patients).

In addition to TNF-α, several studies have shown the importance of IL-17 and IL-12/23 in the pathogenesis of axial Spa [110]. Based on an analysis of pooled data from three previous studies (Measure 1, 2 and 3), Deodar et al. did not show an increased risk of AU in patients taking secukinumab, a humanized monoclonal antibody against IL-17A, for Spa [111]. Other IL-17A inhibitors, such as ixekizumab, have shown promising results [112]. Ustekinumab, a monoclonal antibody, directed to the p40 subunit of IL-12 and IL-23, is approved by the European Medicines Agency (EMA) and the Food and Drugs Administration (FDA) for PsA and is currently under study for uveitis (www.clinicaltrials.gov NCT02911116 and French USTEKINISU trial). 

### 2.3. Behçet’s Disease

#### 2.3.1. Case Report

The case of BD that we present is a 39-year-old male, with a history of smoking and episodes of bilateral acute AU in the last 5 years, treated with topical steroids. No etiology had been demonstrated in previous evaluations. He presented to our emergency consultation with a red and painful left eye for the past 2 days, associated with vision loss. Best corrected visual acuity was 0 LogMAR in the right eye, and 1.0 LogMAR in the left eye. Intraocular pressure was normal and slit lamp examination revealed bilateral acute AU with non-granulomatous keratic precipitates, mild cellular anterior chamber reaction, and no posterior synechiae. Fundus exam revealed mild vitritis, stage 3 bilateral papilledema, venous beading in the posterior pole and periphery, and several patches of perivascular retinal whitening consistent with ischemia. There was a larger yellow-white patch of retinal opacification in the left posterior pole associated with intraretinal hemorrhage consistent with retinal necrosis (Figure 3).

OCT of the right macula found mild intraretinal edema (Figure 4A) and in the left macula there was a large serous retinal detachment, a nodule of intraretinal hyper-reflectivity with loss of differentiation of the inner retinal layers at the site of retinal necrosis, as well as intraretinal edema, and hyperreflective dots in the vitreous (Figure 4B). 

Fluorescein angiography revealed early loss of perfusion at the level of the necrotic nodule in the left eye, and bilateral diffuse capillaritis associated with papillitis (Figure 5A,B). Indocyanine green (ICG) angiography was normal in the right eye and showed several hypofluorescent spots on the middle and late sequences in the posterior pole of the left eye, consistent with patches of choroiditis (Figure 5C,D). 

The patient was addressed on the same day to the internal medicine department. Medical history revealed past episodes of oral ulcers, and general exam found a few eczematous skin lesions, pseudofolliculitis, and was otherwise unremarkable. Laboratory workup showed normal CBC, liver enzymes, kidney function, CRP and angiotensin converting enzyme (ACE). Human immunodeficiency virus (HIV) and syphilis testings were negative, as well as Interferon Gamma Release Assay, Lyme disease and Bartonella. Anterior chamber tap of the left eye for herpesviruses 1 and 2 PCR and universal PCR was negative. HLA typing found no B51 nor B27 haplotype. Chest computed tomography (CT) was unremarkable, but brain MRI found multiple and bilateral subcortical hyperintense T2 spots. Spinal tap found only mild hyperproteinorachia (0.48 g/L). The diagnosis of BD was established according to international criteria given the presence of oral aphtosis and uveitis. 

Initial treatment comprised intravenous acyclovir pending the results of anterior chamber tap. Upon negative results, methylprednisolone pulses were administered over three days followed by slow tapering with oral corticosteroids and treatment with azathioprine and infliximab. Colchicine was also started in prevention of recurring oral ulcers. The ophthalmologic signs rapidly decreased with complete regression of AU, vitritis, macular edema (Figure 5C) and vasculitis at 1 month, and persistence of a mild bilateral papillitis and retinal scarring in place of the necrosis nodule in the left eye at 3 months follow-up (Figure 5D). At last visit 6 month later there had been no relapse in uveitis or oral ulcers.

#### 2.3.2. Ophthalmic Features and Diagnosis

The proportion of patients with BD who experience ocular manifestations varies across studies from 28% to 50% [113], with a higher incidence in men towards the end of the third decade of life [114]. Uveitis can be either inaugural (10–20%) or develop within 2 to 3 years after the onset of oral ulcers. BD represents 4.2% of all uveitis cases in the ULISSE study [29] and 1.8% to 6.1% of severe uveitis treated in European specialized centers [18,19,20]. Among patients with BD’s uveitis, 90% have posterior uveitis or panuveitis and 78% have bilateral involvement; furthermore, 89% have retinal vasculitis and 44% macular edema [114], which can cause visual impairment (with blindness in 13–21% of cases) [64,115]. Visual prognosis is particularly poor in patients with more than three ocular attacks per year, strong vitreous opacity, and exudates within the retinal vascular arcade [116]. Isolated AU is more rarely reported, and mainly affects women (5–10%) [114]. Although hypopyon is considered to be a hallmark of the disease, it has been observed in only 9 to 10.4% of patients in two recent reports [117,118].

The diagnosis of BD has a clinical basis. According to the International Study Group Criteria, in addition to oral ulcers (mandatory criterion), the patient must meet at least two minor criteria (genital ulcers, uveitis, erythema nodosum or papulopustular lesions, and positive pathergy test) and no evidence of differential diagnosis [63]. To compensate for the low sensitivity of these criteria, estimated at 86.2% in a large series of patients (2069 BD and 1519 controls) [66], International Criteria for BD (ICBD) have been developed, revised, then published in 2014 [59]. According to those criteria, a diagnosis of BD is established when a patient scores at least 4 points in the following list of items: oral aphthosis 2 points, genital aphthosis 2 points, ocular manifestations 2 points, skin manifestations (pseudofolliculitis, erythema nodosum, skin aphthosis) 1 point, vascular manifestations 1 point, neurological manifestations 1 point, and positive pathergy test 1 point. A subsequent study in Iran confirmed that the International Criteria for BD was superior to International Study Group Criteria in terms of sensitivity and accuracy [119]. Other clinical and biological features may help orientate the diagnosis of BD. Among them, the presence of HLA B51 allele is a classical biological marker. Nevertheless, the performance of HLA typing in BD in order to evidence the HLA B51 allele suffers from a low sensitivity (51%) with a specificity of 71% [59]. Tugal-Tutkun et al. have recently developed an algorithm for the diagnosis of BD uveitis based on ocular findings only, using retrospective and prospective analysis data from two independent uveitis populations with and without BD [120]. Ten items with a diagnostic odds ratio > 5 were identified. The items with the highest ratios included superficial retinal infiltrate, evidence of occlusive retinal vasculitis, diffuse retinal capillary leakage, and the absence of granulomatous AU or choroiditis in patients with vitritis. These results need to be validated in a larger clinical cohort to assert their performance in different contexts.

#### 2.3.3. Treatment

The treatment of Behçet’s uveitis aims to obtain a complete disappearance of the ocular inflammation. Patients with isolated AU can be treated with topical corticosteroids [121]. According to 2018 EULAR recommendations for the management of BD, systemic immunosuppressive drugs such as azathioprine could be considered if poor prognosis factors are present including young age, early onset, and male gender. The involvement of the posterior segment justifies systemic glucocorticoid therapy in combination with azathioprine. ciclosporin-A is another option to preserve visual acuity and to prevent relapses [122]. Side effects of ciclosporin-A such as renal dysfunction and hypertension should be kept in mind. There is also evidence of a possible association between the use of ciclosporin-A and the development of neurological BD [123]. Methotrexate is also useful and is a possible alternative to azathioprine and ciclosporin-A [62].

However, these treatments are inadequate in patients with severe retinal vasculitis [124]. Acute sight-threatening uveitis and severe uveitis (e.g., severe inflammatory optic neuropathy, macular ischemia, unilateral uveitis in monophthalmic patients) are emergencies for patients with BD. Methylprednisolone pulses (250–1000 mg/d for 1–3 days) are recommended [122]. The prognosis of severe or refractory forms has been radically changed by the introduction of TNF-α antagonists, mainly infliximab [97,125,126]. These drugs are effective in 80% to 90% of cases, usually within a few days [125]. According to the 2018 EULAR recommendations, patients with an initial or recurrent episode of acute sight-threatening uveitis should be treated with infliximab or interferon-α [121]. A recent Spanish study comparing infliximab and ADA in refractory uveitis due to BD showed that both drugs were equally effective, although ADA appeared to be associated with a better outcome in terms of anterior chamber inflammation, vitritis, best-corrected visual acuity and drug retention rate after 1 year of follow-up [127]. Interferon-α2a is also effective in posterior uveitis due to BD, with 2 to 4 weeks delay in onset of action [128]. In contrast to experience with TNF-α antagonists, interferon-α2a induces a sustained remission, which may persist after discontinuation of treatment (20% to 58% of cases) [129], and is associated with less frequent relapses in elderly patients [130]. Yalçindag and Köse recently compared infliximab and interferon in a retrospective study of 53 patients and found no significant difference in the control of intraocular inflammation [131].

Although another study of an IL-1 inhibitor (gevokizumab) for BD uveitis has failed to show efficacy, canakinumab and anakinra appear to be useful alternatives in small groups with BD uveitis, but further trials are needed to demonstrate their efficacy [132,133,134]. Pegylated interferon-α, daclizumab, and secukinumab did not meet the primary endpoints for uveitis compared with placebo in three randomized trials [135,136,137]. Rituximab has shown positive preliminary results, in combination with prednisone and MTX, in a single-blinded trial of 20 patients [138], but sufficiently powered trials are needed to provide evidence of efficacy.

Intravitreal glucocorticoids injections may be an option in patients with unilateral flares as an adjunct to systemic treatment [121]. However, complications are frequent (49%), with cataracts (36%), increased intraocular pressure (43%), and glaucoma (9%) [128]. An algorithm for the management of Behçet’s uveitis reflecting the approach of the French uveitis specialists is shown in Figure 6. In an open-label study, Martin-Varillas et al. showed that optimization of ADA, by progressively increasing the interval between doses, was effective, safe and cost-effective [139]. 

### 2.4. Sarcoidosis

#### 2.4.1. Case Report

The sarcoidosis case presented is that of a 55-year-old woman, self-referred to our ophthalmology clinic in the context of a red and painful left eye associated with a visual loss. The best corrected visual acuity was 0.2 LogMAR in the right eye and 0.4 LogMAR in the left eye. Intraocular pressure was 15 mmHg (9–21 mmHg) in both eyes. Slit lamp examination revealed on the left eye numerous large mutton fat keratic precipitates, predominantly at the inferior pole, with moderate cellular anterior chamber reaction and a posterior synechia (Figure 7). The right eye had sequelae of posterior synechia. No iridal nodule was seen. Fundus examination revealed bilateral mild vitritis with inferior snowballs, cystoid macular edema, and multiple inferior and peripheral waxy spots (Figure 7A) typical of multifocal choroiditis. OCT confirmed the presence of a large cystoid macular edema (Figure 7B) and papilledema. Fluorescein angiography showed progressive filling of macular edema cysts and papillitis (Figure 7A,B), diffuse capillaritis (Figure 7C) and an association of hypofluorescent and hyperfluorescent spots in the inferior periphery, corresponding to choroidal granulomas at different stages of evolution (Figure 7D). ICG angiography revealed multiple choroidal granulomas that were better visualized in early phase at the posterior pole (Figure 7E,F) and at the periphery (Figure 7G,H). OCT-Angiography of the posterior pole at the level of the choriocapillaris revealed areas of low flow co-located with hypofluorescent ICG spots, corresponding to choroidal granulomas. It has been shown recently that OCT-Angiography has a 94% sensitivity for the detection of posterior pole choroidal granulomas [140].

The patient was referred to the internal medicine department. The non-ophthalmological clinical examination was unremarkable. Laboratory testing found normal ESR and CRP, as well as normal serum protein electrophoresis. CBC was normal; Interferon Gamma Release Assay, HIV, Syphilis, Borrelia, and Lyme disease tests were negative. ACE was elevated at 98 U/L (12–68 U/L) and Lysozyme was elevated at 21.3 mg/L (<15 mg/L). Chest CT evidenced bilateral hilar lymphadenopathy, without interstitial infiltrates or pulmonary nodules. Salivary gland biopsy found lymphocytic sialadenitis without granuloma. Transbronchial needle biopsy of lymphadenopathy evidenced multiple noncaseating granulomas compatible with sarcoidosis on two of the four biopsies. Given the severity of the posterior involvement and the significant vision loss, the patient underwent a course of intravenous methylprednisolone pulses over 3 days, followed by slow oral tapering of prednisolone, and topical steroids for the management of AU.

During follow-up, the patient experienced bilateral regression of her AU as well as regression of angiographic and OCT signs. The visual acuity at last visit was 0.2 LogMAR in the right eye and 0.2 LogMAR in the left eye.

#### 2.4.2. Ophthalmic Features and Diagnosis

Sarcoid uveitis is generally bilateral (75–90%) with the same characteristics and clinical course in both eyes [72,141]. AU is usually the most common anatomical form of intraocular inflammation (41–75% of sarcoid uveitis) [72,142,143], followed by posterior, intermediate uveitis, and panuveitis. Nevertheless, recent reports from tertiary centers identified panuveitis as the most commonly encountered subtype of uveitis in sarcoidosis patients [141,144]. AU is usually chronic, bilateral, granulomatous, and associated with anterior and posterior synechiae. Granulomatous AU is classic but not specific to sarcoidosis, as it may be seen in uveitis related to other causes, such as infections (tuberculosis, syphilis, and herpes viruses) and inflammatory disorders (multiple sclerosis and VKH). In contrast, hypopyon is atypical of sarcoidosis-associated uveitis [145]. Chronic anterior sarcoid uveitis can lead to band keratopathy, glaucoma, and cataract formation. This presentation is the paradigm of sarcoid uveitis but, statistically, non-granulomatous uveitis is a more frequent presentation (being twice as frequent in some series) [146]. Indeed, patients with Löfgren’s syndrome most often present with bilateral acute and non-granulomatous AU. In these cases, the inflammation is often self-limiting, as is the systemic disease.

Intermediate uveitis is seen in 6% to 19% of patients with sarcoid uveitis [141,144,146]. On the other hand, sarcoidosis is responsible for 7–18% of all intermediate uveitis cases, which makes it one of the two most important systemic associations along with multiple sclerosis [18,147]. Vitreous opacities may include “snow-balls” and/or “string of pearls” and these features are highly suggestive of a granulomatous process.

In sarcoid posterior uveitis, the retinal lesions usually accompany choroidal inflammation; however, retinal or choroidal involvement can be isolated. Posterior uveitis is observed in 7–28% of patients with ocular sarcoidosis [72,141]. Characteristic findings include retinal periphlebitis associated with segmental cuffing, extensive sheathing, and perivenous infiltrates, referred to as “candle-wax drippings” (70% of posterior sarcoid uveitis cases). These lesions may be subclinical and only visible on fluorescein angiography. Capillary closure and ischemia are uncommon. Peripheral and, sometimes, central multifocal choroiditis is the second hallmark of sarcoidosis at the posterior segment. These lesions are small (less than one-half disk diameter), creamy or white, rarely affecting the macula but predominantly post-equatorial, and more commonly in the inferior fundus. Upon resolution of the granuloma, an area of pigmented epithelial atrophy may occur [71].

Optic-disc nodule(s)/granuloma(s) or a solitary choroidal nodule are rare manifestations, even in sarcoidosis, but are highly specific of the disease [148]. Posterior sarcoid uveitis is associated with neurological involvement in up to 27% of cases [149]. A thorough fundus examination should be performed in all patients suspected of having posterior sarcoid uveitis to differentiate it from optic neuritis. Finally, bilateral papilledema, caused by intracranial hypertension in the context of neuro-sarcoidosis with hydrocephalus, is a classic pitfall since it can mimic features of posterior segment involvement whereas the papilledema is more “mechanical” than inflammatory.

Panuveitis represents 9% to 48% of sarcoid uveitis cases [71,141,144]. Conversely, sarcoidosis is responsible for about 7% to 27% of panuveitis cases and is one of the most frequent etiology in Europe and Japan, along with BD and tuberculous uveitis [18,46].

Physicians should be aware of tattoo-associated uveitis with granulomatous tattoo reaction, with or without systemic sarcoidosis. As shown in the medical literature, this entity becomes more frequent with the increasing popularity of tattooing and affected one patient in our study [150].

Few studies have described long-term systemic outcomes in patients with sarcoid uveitis [144,151,152]. In the most recent studies, 62% to 78.8% of patients had isolated sarcoid uveitis. Of these patients, 7.7% to 32.9% developed extraocular involvement, the most common being in the lungs and skin [144,151]. Han et al. recently reported on a series of 249 patients with uveitis and found that 4 of the 19 patients (21%) with presumed sarcoidosis had ventricular tachycardia requiring cardiac defibrillator implantation [153]. These data are inconsistent with recent studies on patients with sarcoid uveitis, which report a prevalence of cardiac involvement in 1% to 2% of patients [141].

Sarcoid uveitis is associated with a favorable visual outcome, with most patients experiencing mild or no visual impairment [71,151]. However, 2.4% to 10% of patients with sarcoid uveitis develop severe visual impairment [142,144,151]. The main cause of visual loss is cystoid macular edema [151,154]. Poor visual prognosis has been associated with advanced age, African American origin, female sex, underlying chronic systemic disease, posterior ocular segment involvement, multifocal choroiditis, the presence of cystoid macular edema, persistent ocular inflammation, and glaucoma [149,151,152,154].

The evaluation of patients with uveitis and suspected sarcoidosis should begin with non-invasive laboratory and radiological tests and progress to invasive tests if necessary. Increased ACE levels and lymphopenia, as lysozyme, soluble IL-2 receptor and chitotriosidase could be useful for the presumptive diagnosis of sarcoid uveitis (Table 2). The sensitivity of chest-CT for biopsy-proven pulmonary sarcoidosis is higher than that of chest X-ray (91% to 100% versus 41% to 69%, respectively) [142,155,156,157,158,159]. Chest-CT is particularly useful in patients over 50 years of age [160,161,162]. It suggests sarcoidosis if it shows bilateral hilar and/or mediastinal lymph nodes, perilymphatic micronodules, or other parenchymal lung abnormalities [162]. Several studies have demonstrated the value of combining ACE with chest X-ray or chest CT for the diagnosis of sarcoid uveitis [142,161]. A modification of these serum markers and/or the presence of mediastinal lymphadenopathy was observed in almost all patients with histologically-proven sarcoidosis [142]. In a series of 19 patients, we first demonstrated the value of 18-FDG PET to detect occult sites of the disease in patients with unexplained uveitis. As illustrated by our observation, this exam can help determine the most accessible site for biopsy. In a subsequent study of 54 patients with chronic uveitis suggestive of sarcoidosis, we showed that 31% of patients had an increased uptake of 18F-FDG in the mediastinal lymph nodes consistent with sarcoidosis [163]. The diagnostic performance of 18F-FDG PET was higher in elderly patients (especially over 56 years of age), in patients with posterior synechia and in patients with mediastinal lymphadenopathies on the chest CT. More recently, we reported that nearly 30% of our patients with suspected sarcoid uveitis and who had a normal chest CT, had hypermetabolic foci on their 18F-FDG PET consistent with sarcoidosis [164]. This led to a change in diagnostic classification in approximately 21% of patients. This work showed that older age at diagnosis, the presence of synechiae and elevated ACE were significantly associated with abnormal 18F-FDG PET findings. Among invasive tests, a minor salivary gland biopsy is only contributive for the histologic diagnosis of sarcoidosis in uveitis patients with increased serum ACE level and/or compatible imaging findings [165]. Several studies have suggested that bronchoalveolar lavage (BAL) fluid analysis may be useful in the diagnosis of sarcoidosis, even when chest imaging is normal [166,167], but bronchial biopsy are never positive in this situation. The sensitivity of bronchial biopsy in patients with suspected ocular sarcoidosis ranges from 42% to 61% for radiologic stage 0 and from 43% to 84% for stage I [168,169,170]. Endobronchial or endo-esophageal fine needle aspiration of the mediastinal nodes guided by ultrasound may help delay mediastinoscopy [72,161].

Several non-invasive biomarkers may also be useful in diagnosing ocular involvement in sarcoidosis (Table 2). ACE level has been shown to be an efficient biomarker when associated with lymphopenia [43]. These two laboratory tests are available at most centers worldwide. The optimal threshold varies (according to studies with different receiver operating characteristic (ROC) curves) between 52 and 61 UI/l. ACE level benefits from a high negative predictive value (NPV) (from 89% to 97% in patients referred for uveitis), thus making the diagnosis of sarcoidosis very unlikely in patients with uveitis and normal ACE [43,44]. Cotte et al. demonstrated that the presence of lymphopenia can improve the PPV of the test from 45% to 74% with the disadvantage of a less sensitive test with a decreasing sensitivity from 46% to 19%. Test performances were not validated with dedicated studies for ocular sarcoidosis. Sahin et al. demonstrated that elevated serum lysozyme was associated with ocular sarcoidosis but also with other infectious uveitis including tuberculosis and syphilis [47]. Given the similar sensitivity and specificity compared with ACE levels and its interest combined with chest imaging, lysozyme can be useful in diagnosing ocular sarcoidosis. Its usefulness could be theoretically greater in patients treated with ACE inhibitors for whom ACE levels are uninterpretable [46,72]. Those three biomarkers are included in the revised IWOS criteria for the diagnosis of ocular sarcoidosis [171]. Other biomarkers are available for the diagnosis of ocular sarcoidosis such as chitotriosidase activity and serum IL-2 receptor (sIL-2R) levels. Given its poor accessibility, it is difficult to routinely recommend chitotriosidase activity assessment even if its diagnostic performances seems interesting in systemic sarcoidosis [48]. No data is available for the diagnosis of ocular sarcoidosis. Similar accessibility issues are encountered with sIL-2R level assessment. Unlike the chitotriosidase activity assessment, sIL-2R level was evaluated in ocular sarcoidosis. Gundlach et al. reported that sIL-2R measurement had higher sensitivity, specificity and NPV compared with ACE [45]. Accordingly, its Youden index is the highest among sarcoidosis biomarkers (a Youden index near one indicating a highly performant test). However, the revised IWOS criteria are currently excluding sIL2-R measurement on the basis of its poor accessibility [171]. CD4/CD8 ratio in vitreous fluid could also be an interesting biomarker but was considered to be too invasive to be routinely recommended for the diagnosis of ocular sarcoidosis by the IWOS experts.

Based on previous studies on the relevance of laboratory and imaging findings to assess sarcoidosis in uveitis patients, we proposed an algorithm for such patients (Figure 8). Given the high cost of 18F-FDG PET and dosimetric considerations, we believe that its use should be limited to patients with positive predictive factors.

A biopsy is unacceptable for many patients with suspected sarcoidosis and uveitis. Therefore, the first International Workshop on Ocular Sarcoidosis (IWOS) published criteria for the diagnosis of intraocular sarcoidosis eleven years ago [148]. These criteria include a combination of suggestive ophthalmological findings and laboratory investigations when a biopsy is not performed or is negative. The IWOS criteria have been retrospectively validated in Japan with a control group consisting mainly of patients with BD or VKH syndrome [172]. Based on a multicentric retrospective review of medical records, including 884 patients in 12 countries, Acharya et al. showed that the 2009 IWOS clinical criteria and the investigational tests had low sensitivities, except for bilateral hilar lymphadenopathy, and that among 264 patients suspected of having sarcoidosis, 97 (37%) did not met the criteria [173]. To overcome these limitations, the revised IWOS criteria were recently established in an international meeting. The survey and subsequent workshop reached consensus agreements on four criteria which are summarized in Table 3 [171]. The most substantial changes were the addition of 4 systemic investigations: (1) lymphopenia; (2) CD4 alveolar lymphocytosis; (3) parenchymal lung changes consistent with sarcoidosis; and (4) abnormal label uptake on 67-Ga scintigraphy or 18F-FDG PET. A recent Japanese retrospective study showed that the revised IWOS criteria were useful, but could be improved by modifying the criterion of presumed or probable OS [174].

#### 2.4.3. Treatment

The literature concerning the medical treatment of uveitis in patient with sarcoidosis is full of case reports, uncontrolled studies, and small case series [72]. The treatment of sarcoid uveitis largely follows the general principles of idiopathic uveitis (Figure 9) [71]. Almost all patients required local treatment (steroids) while 45% to 70% required systemic treatment, mainly for isolated ocular inflammation [144] and sometimes for concomitant ocular inflammation and active systemic disease [151]. MTX is the most widely used immunosuppressive agent [71]. As with other forms of severe non-infectious uveitis, monoclonal antibodies against TNF-α have been used in sarcoid uveitis [175]. Although one randomized clinical trial with the fusion protein etanercept did not report any improvement [176], several case reports [177] have shown the efficacy of humanized anti-TNF-α monoclonal antibodies for the treatment of refractory ocular sarcoidosis, defined by failure of second-line immunosuppressants to achieve satisfactory disease control [178]. In addition, several case-series have reported the efficacy of infliximab and ADA in ocular sarcoidosis [179,180]. In the same way, we reported the efficacy of TNF-α antagonists in 12 of 18 (67%) patients with refractory sarcoid uveitis [181]. Severe adverse events were frequent, mainly infections, requiring anti-TNF-α interruption in 33% of patients. Overall, TNF-α antagonists appear to be effective in severe and refractory sarcoid uveitis. Their efficacy seems to be temporary and relapses occur in most of these patients within 3 months after treatment discontinuation. In our experience, refractory uveitis in sarcoidosis is unusual (11/301 patients (3%); unpublished data) and clinicians should first rule out non-compliance, infectious uveitis, or lymphoma before starting infliximab or ADA [182,183]. Novel therapeutic approaches have been investigated in refractory sarcoidosis with biologics such as rituximab, tocilizumab, or small molecules, including JAK inhibitors. The medical literature supports their use in refractory sarcoid uveitis in case reports [184,185,186,187].

## 3. Discussion

In our tertiary center, among 253 patients referred by ophthalmologists for the diagnostic work-up of uveitis in the last two years, 75 (29.6%) have been diagnosed with Spa, BD, or sarcoidosis. Other diagnosed conditions included HLA-B27 related uveitis (7.9%), VKH (1.5%), multiple sclerosis (1.5%), tubulointerstitial nephritis with uveitis (0.7%), other inflammatory diseases (1.1%), infectious uveitis (8.5%), purely ophthalmological entities (1.6%), lymphoma (2.6%), and uveitis related to drug side effects (1.1%). No underlying condition was found in 111 of 253 (44.1%) patients. In all nine patients with BD, the presence of oral aphtosis together with genital ulcers or skin lesions was sufficient for the diagnosis, which had not been considered before. Similarly, 20 patients with Spa presented unexplained chronic back pain that led to HLA-B27 determination. Among these patients and in case of normal radiograph of the pelvis a sacroiliac MRI was performed [90]. The most frequently identified systemic disease was sarcoidosis (17.2%, *n* = 46), with 22 histologically proven cases on biopsy: 7 from bronchial samples, 3 from mediastinal lymph nodes, 5 from minor salivary glands, 3 from skin, and 1 from each of peripheral lymph node, thyroid, and stomach. Three patients with normal bronchial samples and normal minor salivary gland biopsy had endobronchial ultrasound-transbronchial needle aspiration of intrathoracic nodes that showed granuloma. Based on Abad’s modified criteria, 17 patients had presumed sarcoid uveitis on the basis of either elevated ACE and positive chest CT (*n* = 11) or positive 18F-FDG PET suggestive of sarcoidosis (*n* = 4) and both positive chest CT and 18F-FDG PET (*n* = 2), while 9 patients had possible sarcoidosis (according to the American Thoracic Society guidelines) on the basis of isolated positive 18F-FDG PET (*n* = 7). Only 10 of the 46 patients had an abnormal chest X-ray which was suggestive of sarcoidosis.

To our knowledge, only limited data is available in the medical literature on how the medical history, physical examination, laboratory, and imaging investigations can be useful to make a diagnosis in previously described clinical entities. In the ULISSE study, Parisot et al. found that an etiological diagnosis of uveitis was established for 75.7% of patients within the first step of the standardized strategy which included a minimal work-up with non-expensive laboratory investigations (CBC, ESR, CRP, tuberculin skin test, syphilis serology, and chest X-ray) and extra diagnostic tests guided by clinical or paraclinical findings. Another 22.8% of cases were diagnosed after the second or third step which included more complex investigations according to ophthalmological findings. At the end of the third step, extra investigations at the physician’s discretion enabled a diagnosis in the remaining cases [29]. Although vitreous measurement of CD4/CD8 ratio seems invasive, one can assume that, in the near future, measures of CD4/CD8 ratios of cellular infiltrates in aqueous humor or cytokine levels in aqueous humor, tears or serum could help diagnose these three inflammatory disorders and stratify patients for tailored treatments [188,189,190].

Choi et al. recently reported their experience on 179 patients referred to their center with a diagnosis of idiopathic uveitis. They were able to establish a diagnosis in 52 patients (29%). Sarcoidosis was the most common diagnosis (*n* = 19, 36.5%), followed by HLA-B27-associated uveitis while one patient had BD, and no underlying condition was found in 127 of 179 patients (70.9%) [191].

Our results underscore the need for rapid diagnosis in these fatal pathologies, and for a multidisciplinary approach, in parallel with evaluation by the ophthalmologist. 

## Figures and Tables

**Figure 1 jcm-10-00281-f001:**
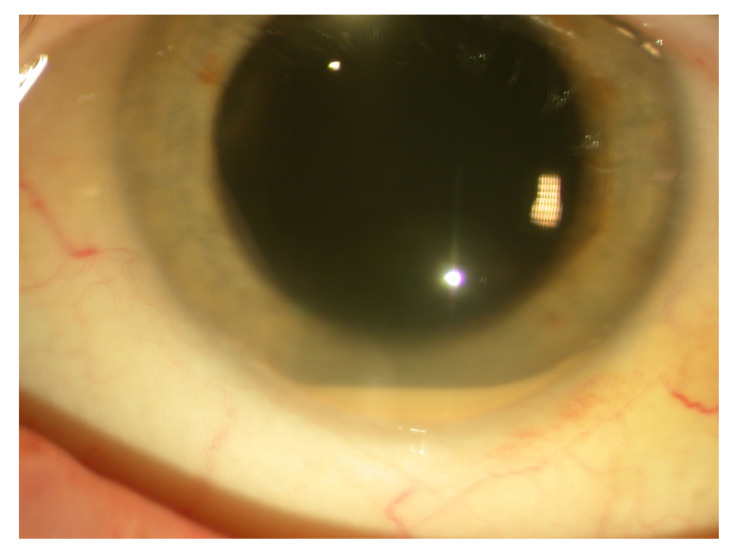
Slit lamp examination of case 1 showing nongranulomatous corneal precipitates with a cellular anterior chamber reaction and hypopyon.

**Figure 2 jcm-10-00281-f002:**
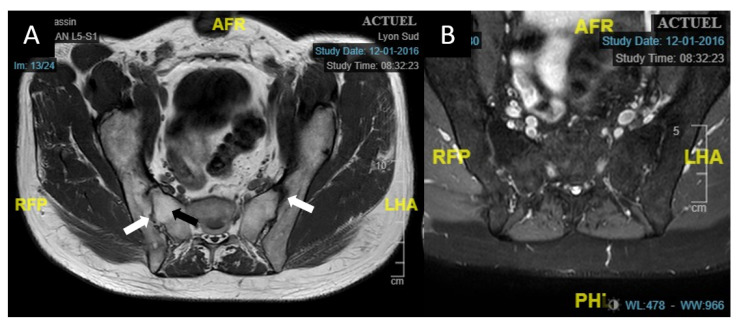
Axial sacroiliac MRI of case 1. (**A**): T1-weighted sacroiliac axial MRI of case 1 showing major joint reshaping in the form of hypo dense areas bordering the joint (white arrows) as well as ankylosis of the right sacroiliac joint with a T1 hypersignal of the subchondral bone. (**B**): T2 SPAIR axial sequence without hypersignal of the sacroiliac joints, indicating the absence of active involvement. Abbreviations: A = anterior, P= posterior, R = right, L = left, H = head, F = foot; MRI: magnetic resonance imaging; T2 SPAIR: T2 Spectral Attenuated Inversion Recovery.

**Figure 3 jcm-10-00281-f003:**
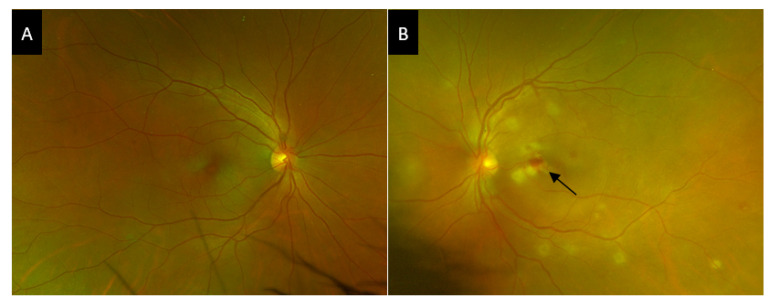
Wide field fundus photograph of the right eye of the case 2 patient. (**A**), Widefield fundus photograph of the right eye showing no signs of intermediate or posterior uveitis. (**B**), Widefield fundus photograph of the left eye showing multiple patchy subretinal white spots and a nodule of retinal necrosis with hemorrhage (black arrow).

**Figure 4 jcm-10-00281-f004:**
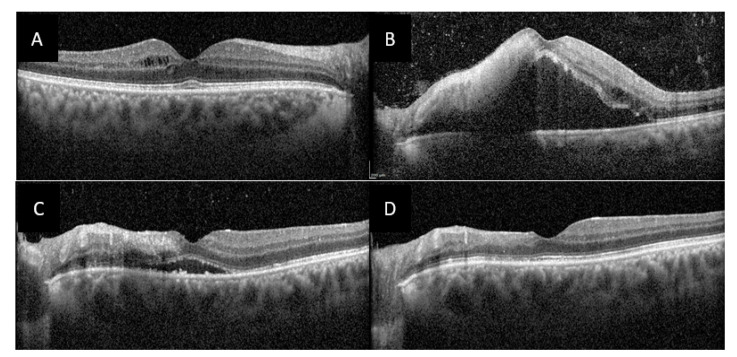
Optical coherence tomography (OCT) of both macula of case 2 patient. (**A**), OCT of the right macula showing a few cysts of intraretinal edema. (**B**), OCT of the left macula showing a large serous retinal detachment, a hyperreflective nodule in the inner retinal layers, intraretinal edema and numerous hyperreflective dots in the vitreous. (**C**), OCT of the left macula at 1-month follow-up showing partial regression of subretinal fluid. (**D**) OCT of the left macula at 3 months follow-up showing complete regression of macular edema, and atrophy of internal retina in place of the hyperreflective nodule.

**Figure 5 jcm-10-00281-f005:**
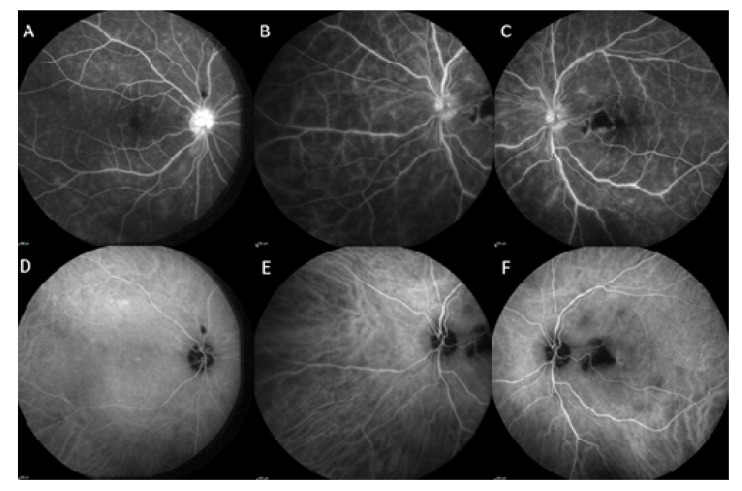
Fluorescein angiogram of both eyes of case 2 patient. Fluorescein angiogram of the right eye (**A**) in late sequence showing diffuse capillaritis and papillitis. Fluorescein angiogram of the left eye (**B**,**C**) in the same sequence showing central and peripheral capillaritis, papillitis, and a mask effect due to hemorrhage in a retinal necrosis nodule. Indocyanine green (ICG) angiogram in middle sequence of the right eye (**D**) showing no choroidal involvement, and of the left eye (**E**,**F**) showing predominantly macular spots of hypofluorescence corresponding to choroiditis.

**Figure 6 jcm-10-00281-f006:**
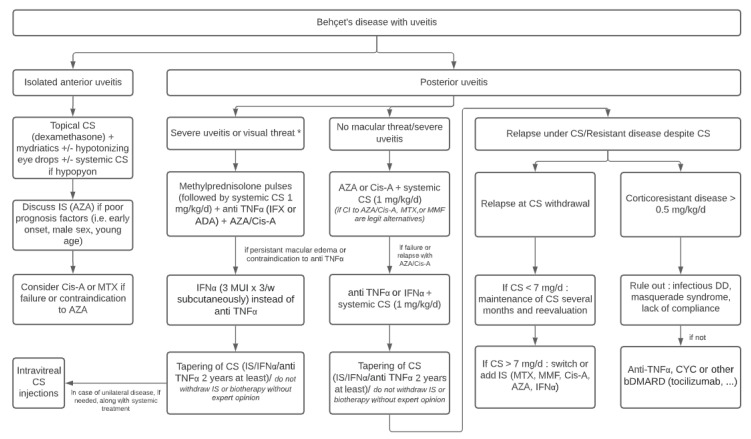
Algorithm for management of Behçet’s uveitis. Abbreviations: ADA: adalimumab; AZA: azathioprine; bDMARD: biologic disease modifying anti rheumatic drugs; Cis-A: ciclosporine A; CS: corticosteroids; CYC: cyclophosphamide; IFN: interferon; IFX: infliximab; IS: immunosuppressants; MMF: mycophenolate mofetil; MTX: methotrexate; MUI: million international units; TNF: tumor necrosis factor. * optic neuropathy, macular edema with visual acuity <20/200 or vasculitis with retinal ischemia.

**Figure 7 jcm-10-00281-f007:**
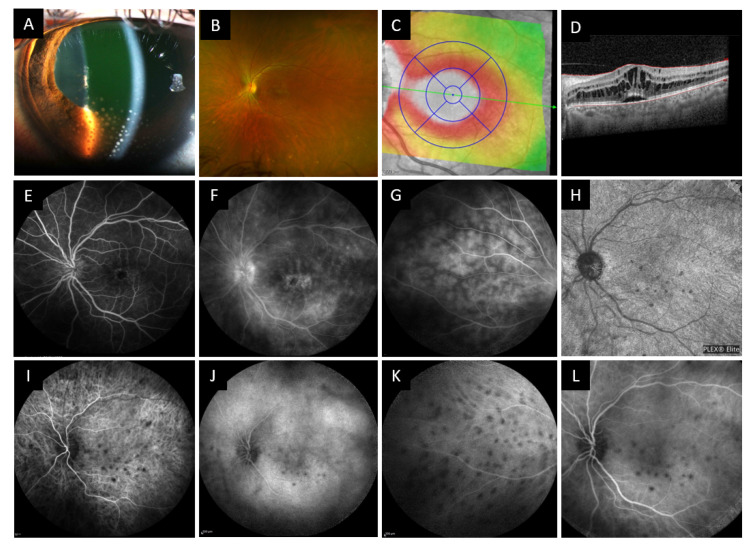
Slit lamp examination, wide field fundus photography, OCT, fluorescein angiography and OCT angiography of case 3 patient. (**A**) Slit lamp examination showing numerous mutton fat keratic precipitates with anterior chamber reaction and posterior synechiae. (**B**) Wide field fundus photography showing macular edema and inferior yellow-white waxy spots. (**C**) OCT Thickness mapping of the macula and B-scan (**D**) showing cystoid macular edema with retro foveolar subretinal detachment. (**E**–**G**) Fluorescein angiography showing signs of cystoid macular edema, papillitis, capillaritis and choroidal granulomas. (**I**–**K**) Indocyanine angiography showing hypofluorescent spots most visible in the early phase, present at the posterior pole and periphery. (**H**) OCT angiography of the choriocapillaris at diagnosis showing spots of reduced choriocapillary flow corresponding to choroidal granulomas on the ICG angiogram (**L**).

**Figure 8 jcm-10-00281-f008:**
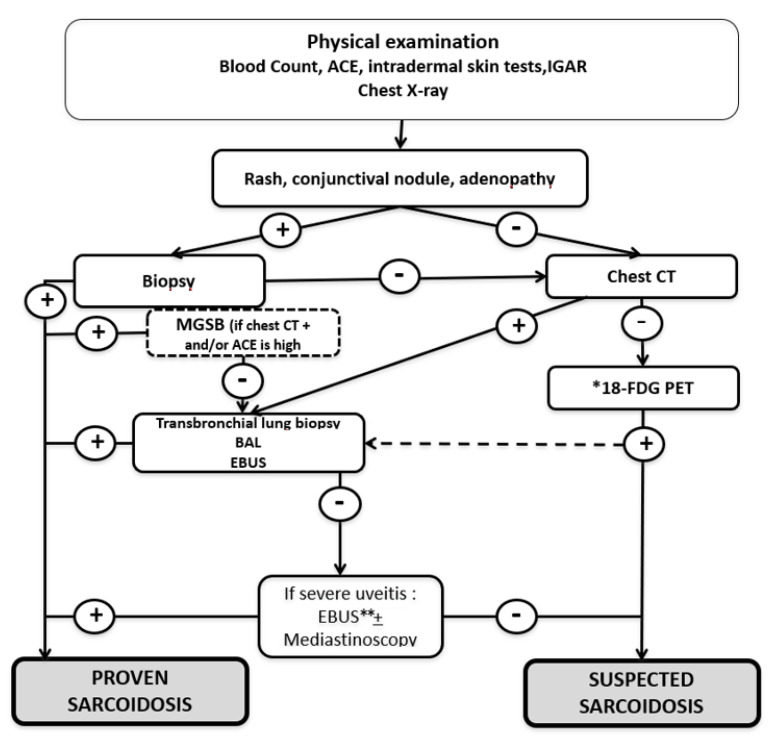
Algorithm to assess uveitis in patients with suspected sarcoidosis (From Sève et al., *Sem Resp Crit Care Med*, 2020). Abbreviations and notes: ACE: angiotensin converting enzyme; BAL: bronchoalveolar lavage; EBUS: endoscopic ultrasound-guided fine-needle aspiration of intrathoracic nodes; MSGB: minor salivary-gland biopsy; 18F-FDG PET: 18-fluorodeoxyglucose positron-emission tomography. * if old age at uveitis presentation, presence of synechia and an elevated ACE; ** If EBUS not previously performed. The yield of diagnoses from a bronchoalveolar lavage and a trans-bronchoscopic biopsy in patients with an abnormal PET and normal chest CT is unknown. Negative mediastinal lymph-node biopsies from patients with mediastinal lymphadenopathy are, in our experience, exceptional.

**Figure 9 jcm-10-00281-f009:**
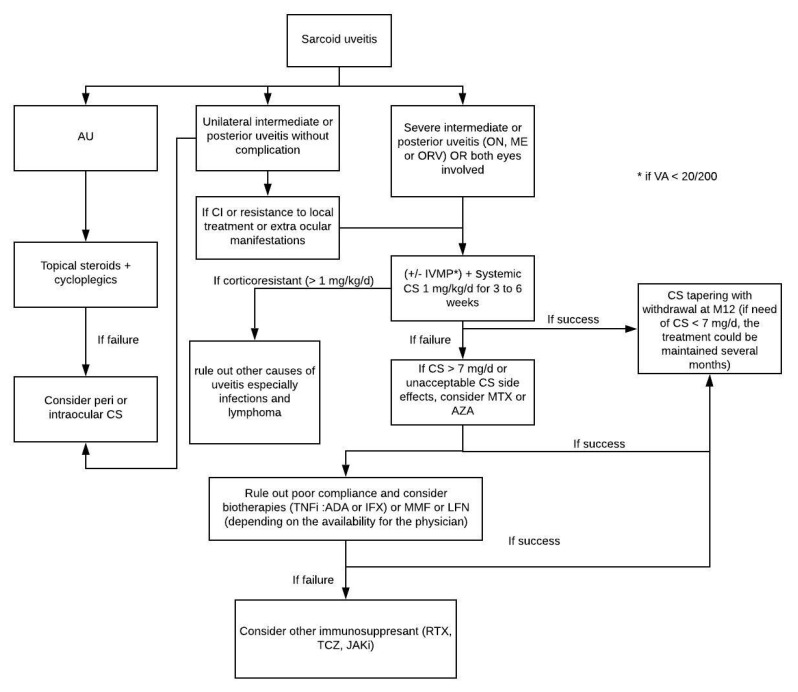
Corticosteroids management in non-infectious uveitis (Sève et al., *Sem Resp Crit Care Med*, 2020). Abbreviations: AU: anterior uveitis; CS: corticosteroids; ON: optic neuritis; ME: macular edema; ORV: occlusive retinal vasculitis; VA: visual acuity; IVMP: intravenous methylprednisolone pulse; CI: contra indication; TNFi: Tumor Necrosis Factor inhibitor; ADA: adalimumab; IFX: infliximab; MTX: methotrexate; AZA: azathioprine; MMF: mycophenolate mofetil; LFN: leflunomide; RTX: rituximab; TCZ: tocilizumab; JAKi: Janus kinase inhibitor.

**Table 3 jcm-10-00281-t003:** Revised criteria of International Workshop on Ocular Sarcoidosis (IWOS) for the diagnosis of ocular sarcoidosis (from [171]).

**I.** **Other causes of granulomatous uveitis must be ruled out**
**II.** **Intraocular signs suggestive of ocular sarcoidosis**
1. Mutton-fat keratic precipitates(large or small) and/or iris nodules at pupillary margin (Koeppe) or in stroma (Busacca)
2. Trabecular meshwork nodules and/or tent-shaped peripheral anterior synechia
3. Snowballs/strings of pearls vitreous opacities
4. Multiple chorioretinal peripheral lesions (active and/or atrophic)
5. Nodular and/or segmental periphlebitis (±candle-wax drippings) and/or macroaneurysm in an inflamed eye
6. Optic-disc nodule(s)/granuloma(s) and/or solitary choroidal nodule
7. Bilaterality (assessed by ophthalmological examination including ocular imaging showing subclinical inflammation)
**III.** **Systemic investigation results in suspected ocular sarcoidosis**
1. Bilateral hilar lymphadenopathy on chest X-ray and/or chest computed CT scan
2. Negative tuberculin test in a BCG-vaccinated patient or interferon-gamma releasing assays
3. Elevated serum angiotensin converting-enzyme
4. Elevated serum lysozyme
5. Elevated CD4/CD8 ratio (>3.5) in bronchoalvelar lavage fluid
6. Abnormal label uptake on 67-gallium scintigraphy or 18F-fluorodesoxyglucose positron emission tomography imaging
7. Lymphopenia
8. Parenchymal lung changes consistent with sarcoidosis, as determined by pneumologists or radiologists
**Diagnostic criteria of ocular sarcoidosis**
Diagnostic criteria of ocular sarcoidosis were established in 3 levels of certainty:
Definite ocular sarcoidosis: diagnosis supported by biopsy with compatible uveitis
Presumed ocular sarcoidosis: diagnosis not supported by biopsy, but bilateral hilar lymphadenopathy present with two intraocular signs
Probable ocular sarcoidosis: diagnosis not supported by biopsy and bilateral hilar lymphadenopathy absent, but three intraocular signs and two systemic investigations selected from two to eight are present

## Data Availability

The data presented in this study are available on request from the corresponding author. The data are not publicly available due to privacy concerns.

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
