# Peer review of "Uveitis as an Open Window to Systemic Inflammatory Diseases"

_jcm, 2021, doi:10.3390/jcm10020281_

Round 1

Reviewer 1 Report

Overall, the manuscript is clearly written and represents some contribution to the field.

The authors should mention biomarkers in patients with uveitis (For example, ACE and soluble IL-2 receptor for sarcoidosis). Summary figure of how biomarkers are useful in uveitis. I understand that this is a complicated topic and agree that this would help the reader.

Author Response

Q : Overall, the manuscript is clearly written and represents some contribution to the field.

The authors should mention biomarkers in patients with uveitis (For example, ACE and soluble IL-2 receptor for sarcoidosis). Summary figure of how biomarkers are useful in uveitis. I understand that this is a complicated topic and agree that this would help the reader.

A : The authors would like to thank the reviewer for his/her reading of our work.

In order to provide more information about biomarkers in the diagnosis of ocular sarcoidosis, we provided an addenda and a supplementary table (table 2). We have chosen to mention the following biomarkers : lymphopenia, angiotensin converting enzyme (ACE), soluble interleukin 2 receptor (sIL-2R), lysozyme and chitotriosidase.

You will find the modifications from line 570 to line 596 : “Several non-invasive biomarkers […] the IWOS experts.” and at line 541 : “as lysozyme, soluble IL-2 receptor and chitotriosidase could be”.

Reviewer 2 Report

The review “Uveitis as an open window to systemic inflammatory diseases” comes from a French research group in Lyon that has previously published several important contributions in the field of uveitis. The authors describe their experience with 75 patients with systemic inflammatory diseases (20 with spondyloarthritis, 9 with Behçet’s disease, and 46 with sarcoidosis), that they observed in the last two years.  Overall, the manuscript is well written and accompanied by clear and demonstrative figures as well as by useful algorithms and tables, and is therefore worthy of publication. However, it results somewhat “heavy” to read because of its length.

The following revisions are suggested:

  • Abstract, lines 25-27: “chest computed tomography (CT) and 18F-fluorodeoxy- glucose positron emission tomography CT that helped to identify smaller hilar or mediastinal involvement”: these features, taken alone, are not diagnostic in themselves but were the reason to further explore those patients.
  • Figure 1 is for students of medicine, and can be deleted.
  • Lines 91-92: “24.5%” is not a universal percentage. It would be better to write “…and affects approximately one forth of the Spa patients”.
  • Lines 119-122: after stating that sarcoidosis is a multi-systemic disease, the sentence can be terminated with “…non-caseating granulomas”, thus deleting the long list of the organs that can be potentially involved.
  • The handling of each of the three systemic inflammatory diseases is preceded by a case report, that is not strictly essential in the context of a review. But if the authors wish to leave this unchanged, it is suggested that at the end of the description of each case report a short comment be added to explain why they chose that case and not another.
  • In the authors’ intent to be all-inclusive, the treatment section of each inflammatory condition results somewhat redundant due to the inclusion of therapeutic attempts whose usefulness is far from being demonstrated. For example, the sentences “Mugheddu et al….” (lines 296-299), “There are also…” (lines 429-431) might be safely deleted with no consequences for the completeness of information to the reader.
  • Lines 525-526: please, explain why “bilateral papilledema, caused by intracranial hypertension in the context of neuro-sarcoidosis, is a classic pitfall”.
  • Since the revised criteria for the diagnosis of ocular sarcoidosis are clearly reported in Table 3, the text corresponding to lines 592-605 can be suitably shortened.
  • Lines 669-670: the statement “9 patients had possible sarcoidosis on the basis of isolated positive 18F-FDG PET (n=7)”, although mitigated by “possible”, remains a leap of faith rather than a diagnosis.
  • 224(!) references are objectively too many and can be reasonably reduced. In spite of this large number, this reviewer feels that, in step with their reduction, the following references (the first for spondyloarthritis, the second and third for Behçet’s disease and the fourth for sarcoidosis) should be added:
  • Bengtsson K, Forsblad-d'Elia H, Deminger A, Klingberg E, Dehlin M, Exarchou S, Lindström U, Askling J, Jacobsson LTH. Incidence of extra-articular manifestations in ankylosing spondylitis, psoriatic arthritis and undifferentiated spondyloarthritis: results from a national register-based cohort study. Rheumatology (Oxford). 2020 Nov 20:keaa692.
  • Abd El Latif E, Abdel Kader Fouly Galal M, Tawfik MA, Elmoddather M, Nooreldin A, Shamselden Yousef H. Pattern of Uveitis Associated with Behçet's Disease in an Egyptian Cohort. Clin Ophthalmol. 2020 Nov 20;14:4005-4014.
  • Tugal-Tutkun I, Çakar Özdal P. Behçet's disease uveitis: is there a need for new emerging drugs? Expert Opin Emerg Drugs. 2020 Nov 30:1-17.
  • Dammacco R, Biswas J, Kivelä TT, Zito FA, Leone P, Mavilio A, Sisto D, Alessio G, Dammacco F. Ocular sarcoidosis: clinical experience and recent pathogenetic and therapeutic advancements. Int Ophthalmol. 2020 Dec;40(12):3453-3467. 
  • The manuscript is written correctly and very few typos have been detected: line 407 “patients)are”; line 425 “did not meet”; line 453 “a posterior synechiae”; figure 9:”conjonctival”

Author Response

Q : The review “Uveitis as an open window to systemic inflammatory diseases” comes from a French research group in Lyon that has previously published several important contributions in the field of uveitis. The authors describe their experience with 75 patients with systemic inflammatory diseases (20 with spondyloarthritis, 9 with Behçet’s disease, and 46 with sarcoidosis), that they observed in the last two years.  Overall, the manuscript is well written and accompanied by clear and demonstrative figures as well as by useful algorithms and tables, and is therefore worthy of publication. However, it results somewhat “heavy” to read because of its length.

A : The authors thanks the reviewer for his/her attentive reading and suggestions.

Q : The following revisions are suggested:

Abstract, lines 25-27: “chest computed tomography (CT) and 18F-fluorodeoxy- glucose positron emission tomography CT that helped to identify smaller hilar or mediastinal involvement”: these features, taken alone, are not diagnostic in themselves but were the reason to further explore those patients.

A : In order to clarify this statement, we added “and allowed to further investigate those patients”. The modification can be found on line 27.

Q : Figure 1 is for students of medicine, and can be deleted.

A : We do agree that this figure can be easily deleted. This element was removed from the final manuscript.

Q : Lines 91-92: “24.5%” is not a universal percentage. It would be better to write “…and affects approximately one fourth of the Spa patients”.

A : We do agree that this statement would be clearer with this formulation. Please find the modifications on line 81 : “and affects approximately one fourth of the Spa patients”.

Q : Lines 119-122: after stating that sarcoidosis is a multi-systemic disease, the sentence can be terminated with “…non-caseating granulomas”, thus deleting the long list of the organs that can be potentially involved.

A : We do agree that the sentence would be easier to read this way. As the reviewer suggested , we have deleted the list of involved organ and stopped the sentence at “non caseating granulomas”. Please find the modification on line 112.

Q : The handling of each of the three systemic inflammatory diseases is preceded by a case report, that is not strictly essential in the context of a review. But if the authors wish to leave this unchanged, it is suggested that at the end of the description of each case report a short comment be added to explain why they chose that case and not another.

A : We would like to clarify that this article should be considered as an original article and not as a typical review. Nevertheless, we clarified the reason we chose these cases on line 143-146 “The case reports were chosen […] diagnostic issues faced by the clinician.”

Q : In the authors’ intent to be all-inclusive, the treatment section of each inflammatory condition results somewhat redundant due to the inclusion of therapeutic attempts whose usefulness is far from being demonstrated. For example, the sentences “Mugheddu et al….” (lines 296-299), “There are also…” (lines 429-431) might be safely deleted with no consequences for the completeness of information to the reader.

A : We agree that these sentences can be deleted without compromising essential information for the reader. These sentences have been deleted.

Q : Lines 525-526: please, explain why “bilateral papilledema, caused by intracranial hypertension in the context of neuro-sarcoidosis, is a classic pitfall”.

A : We do agree that this statement has to be clarified. We modified the sentence : “Finally, bilateral papilledema, caused by intracranial hypertension in the context of neuro-sarcoidosis with hydrocephalus, is a classic pitfall since it can mimic features of posterior segment involvement whereas the papilledema is more “mechanical” than inflammatory” from line 517 to line 518.

Q : Since the revised criteria for the diagnosis of ocular sarcoidosis are clearly reported in Table 3, the text corresponding to lines 592-605 can be suitably shortened.

A : The section has been shortened with the modification “which are summarized in table 3” on line 614.

Q : Lines 669-670: the statement “9 patients had possible sarcoidosis on the basis of isolated positive 18F-FDG PET (n=7)”, although mitigated by “possible”, remains a leap of faith rather than a diagnosis.

A : We clarified this statement as the word “possible” refers to “possible sarcoidosis” as mentioned in the American Thoracic Society guidelines regarding sarcoidosis diagnosis. Thus, we added “(according to the American Thoracic Society guidelines)” on lines 683-4.

Q : 224(!) references are objectively too many and can be reasonably reduced. In spite of this large number, this reviewer feels that, in step with their reduction, the following references (the first for spondyloarthritis, the second and third for Behçet’s disease and the fourth for sarcoidosis) should be added:

Bengtsson K, Forsblad-d'Elia H, Deminger A, Klingberg E, Dehlin M, Exarchou S, Lindström U, Askling J, Jacobsson LTH. Incidence of extra-articular manifestations in ankylosing spondylitis, psoriatic arthritis and undifferentiated spondyloarthritis: results from a national register-based cohort study. Rheumatology (Oxford). 2020 Nov 20:keaa692.

Abd El Latif E, Abdel Kader Fouly Galal M, Tawfik MA, Elmoddather M, Nooreldin A, Shamselden Yousef H. Pattern of Uveitis Associated with Behçet's Disease in an Egyptian Cohort. Clin Ophthalmol. 2020 Nov 20;14:4005-4014.

Tugal-Tutkun I, Çakar Özdal P. Behçet's disease uveitis: is there a need for new emerging drugs? Expert Opin Emerg Drugs. 2020 Nov 30:1-17.

Dammacco R, Biswas J, Kivelä TT, Zito FA, Leone P, Mavilio A, Sisto D, Alessio G, Dammacco F. Ocular sarcoidosis: clinical experience and recent pathogenetic and therapeutic advancements. Int Ophthalmol. 2020 Dec;40(12):3453-3467.

A : We do agree that several references can be easily deleted. We suppressed 35 references and added the four suggested by the reviewer. The final number of references is now 193.

Q : The manuscript is written correctly and very few typos have been detected: line 407 “patients)are”; line 425 “did not meet”; line 453 “a posterior synechiae”; figure 9:”conjonctival”.

Typos have been corrected respectively on lines 401, 419, 444 and in figure 8.

Reviewer 3 Report

This is a well written review that in principle uses three selected case reports to eloborate on clinical presentation, diagnostic work-up and treatment of uveitis in relation to spondyloarthritis, Behçet's disease (BD) and sarcoidosis. In general the review tends to be complete. Yet I do have some minor comments that need to be adressed by the authors.

1). The association between BD and HLA-B51 is not super strong (prevalence of HLA-B51 positivity BD is ~55-60%, compared to 10-20% in general population), indicating that still a subtantial number of BD patients will be HLA-B51 negative. Nevertheless, I would prefer that the authors, under the heading ophthalmic features and diagnosis (2.3.2.), do introduce the association between BD and HLA-B51. Especially since in the diagnostic work-up of their own case patient presented HLA-B51 was determined, and turned out negative.

2). Evidence is accumulating that serum soluble IL2-Receptor (sIL2R) measurement is a useful diagnostic marker for sarcoidosis, including ocular sarcoidosis. There are even indications that sIL2R outperforms ACE in this regard. The IWOS did not include serum sIL2R measurement in their 2019 revised criteria for ocular sarcoidosis (main reason being that it was considered that serum sIL2R measurement was not (yet) used widely enough in uveitis clinics). However, sIL2R measurement is used througout the world in the diagnostic work-up of (ocular) sarcoidosis. Therefore, under the heading ophthalmic features and diagnosis (2.4.2.) the authors should implement text on serum sIL2R measurement in relation to diagnosing ocular sarcoidosis.       

Author Response

Q : This is a well written review that in principle uses three selected case reports to elaborate on clinical presentation, diagnostic work-up and treatment of uveitis in relation to spondyloarthritis, Behçet's disease (BD) and sarcoidosis. In general, the review tends to be complete. Yet I do have some minor comments that need to be addressed by the authors.

A : the authors thank the reviewer for his/her comments and careful reading.

Q : 1). The association between BD and HLA-B51 is not super strong (prevalence of HLA-B51 positivity BD is ~55-60%, compared to 10-20% in general population), indicating that still a substantial number of BD patients will be HLA-B51 negative. Nevertheless, I would prefer that the authors, under the heading ophthalmic features and diagnosis (2.3.2.), do introduce the association between BD and HLA-B51. Especially since in the diagnostic work-up of their own case patient presented HLA-B51 was determined, and turned out negative.

A : A short statement about sensitivity and specificity of HLA-B51 from line 377 to line 381 : “Other clinical and biological features may help to orientate the diagnosis of BD. Among them, the presence of HLA B51 allele in HLA typing is a classical biological marker in BD. Nevertheless, the performance of HLA typing in BD in order to evidence the HLA B51 allele suffers from a low sensitivity (51%) with a specificity of 71% [60].”

Q : 2). Evidence is accumulating that serum soluble IL2-Receptor (sIL2R) measurement is a useful diagnostic marker for sarcoidosis, including ocular sarcoidosis. There are even indications that sIL2R outperforms ACE in this regard. The IWOS did not include serum sIL2R measurement in their 2019 revised criteria for ocular sarcoidosis (main reason being that it was considered that serum sIL2R measurement was not (yet) used widely enough in uveitis clinics). However, sIL2R measurement is used throughout the world in the diagnostic work-up of (ocular) sarcoidosis. Therefore, under the heading ophthalmic features and diagnosis (2.4.2.) the authors should implement text on serum sIL2R measurement in relation to diagnosing ocular sarcoidosis. 

A : The same kind of comment has been made by reviewer 1. We do agree that this modification would be valuable for the manuscript. Please find the modifications about the use of biomarkers and especially on sIL2-R from line 570 to line 596 : “Several non-invasive biomarkers […] the IWOS experts.”